# Performance Evaluation of a Fully Automated Molecular Diagnostic System for Multiplex Detection of SARS-CoV-2, Influenza A/B Viruses, and Respiratory Syncytial Virus

**DOI:** 10.3390/diagnostics15141791

**Published:** 2025-07-16

**Authors:** James G. Komu, Dulamjav Jamsransuren, Sachiko Matsuda, Haruko Ogawa, Yohei Takeda

**Affiliations:** 1Graduate School of Animal and Veterinary Sciences and Agriculture, Obihiro University of Agriculture and Veterinary Medicine, 2-11 Inada, Obihiro 080-8555, Hokkaido, Japan; jkomu@jkuat.ac.ke; 2Department of Medical Laboratory Sciences, College of Health Sciences, Jomo Kenyatta University of Agriculture and Technology, Nairobi P.O. Box 62000-00200, Kenya; 3Research Center for Global Agromedicine, Obihiro University of Agriculture and Veterinary Medicine, 2-11 Inada, Obihiro 080-8555, Hokkaido, Japan; jduuya@obihiro.ac.jp (D.J.); chakachaka0810@gmail.com (S.M.); 4Department of Veterinary Medicine, Obihiro University of Agriculture and Veterinary Medicine, 2-11 Inada, Obihiro 080-8555, Hokkaido, Japan; hogawa@obihiro.ac.jp

**Keywords:** multiplex RT-qPCR, automation, sample-to-result, SARS-CoV-2, influenza viruses, RSV

## Abstract

**Background/Objectives**: Concurrent outbreaks of severe acute respiratory syndrome coronavirus 2 (SARS-CoV-2), influenza A and B viruses (IAV/IBV), and respiratory syncytial virus (RSV) necessitate rapid and precise differential laboratory diagnostic methods. This study aimed to evaluate the multiplex molecular diagnostic performance of the geneLEAD VIII system (Precision System Science Co., Ltd., Matsudo, Japan), a fully automated sample-to-result precision instrument, in conjunction with the VIASURE SARS-CoV-2, Flu & RSV Real Time PCR Detection Kit (CerTest Biotec, S.L., Zaragoza, Spain). **Methods**: The specific detection capabilities of SARS-CoV-2, IAV/IBV, and RSV genes were evaluated using virus-spiked saliva and nasal swab samples. Using saliva samples, the viral titer detection limits of geneLEAD/VIASURE and manual referent singleplex RT-qPCR assays were compared. The performance of geneLEAD/VIASURE in analyzing single- and multiple-infection models was scrutinized. The concordance between the geneLEAD/VIASURE and the manual assays was assessed. **Results**: The geneLEAD/VIASURE successfully detected all the virus genes in the saliva and nasal swab samples despite some differences in the Ct values. The viral titer detection limits in the saliva samples for SARS-CoV-2, IAV, IBV, and RSV using geneLEAD/VIASURE were 10^0^, ≤10^−2^, 10^0^, and 10^2^ TCID_50_/mL, respectively, compared to ≤10^−1^, ≤10^0^, ≤10^0^, and ≤10^4^ TCID_50_/mL, respectively, in the manual assays. geneLEAD/VIASURE yielded similar Ct values in the single- and multiple-infection models, with some exceptions noted in the triple-infection models when low titers of RSV were spiked with high titers of the other viruses. The concordance between geneLEAD/VIASURE and the manual assays was high, with Pearson’s R^2^ values of 0.90, 0.85, 0.92, and 0.95 for SARS-CoV-2, IAV, IBV, and RSV, respectively. **Conclusions**: geneLEAD/VIASURE is a reliable diagnostic tool for detecting SARS-CoV-2, IAV/IBV, and RSV in single- and multiple-infection scenarios.

## 1. Introduction

It has been over 5 years since the initial report of the severe acute respiratory syndrome coronavirus 2 (SARS-CoV-2) infection, which precipitated the coronavirus disease 2019 pandemic. During the pandemic’s early stages, intensified public health measures against SARS-CoV-2 inadvertently resulted in a significant decrease in the circulation of traditional endemic respiratory viruses, including influenza A and B viruses (IAV and IBV) and respiratory syncytial virus (RSV) [1,2]. However, as these restrictions were relaxed, the cocirculation of SARS-CoV-2 with these viruses led to co-infections with severe outcomes [3,4]. Notably, since the winter season of 2022–2023, there have been instances of “tripledemics” due to concurrent outbreaks of SARS-CoV-2, influenza viruses, and RSV in various regions worldwide [5,6,7], causing substantial strain on healthcare systems [8,9].

The overlapping clinical manifestations of diseases caused by SARS-CoV-2, influenza viruses, and RSV complicate the reliance on clinical diagnosis alone for their differentiation. Consequently, the necessity for rapid and accurate differential laboratory diagnosis becomes paramount for the specific therapeutic and public health management of each infection. The primary method for diagnosing SARS-CoV-2, influenza viruses, and RSV infections has been singleplex real-time reverse transcription polymerase chain reaction (RT-qPCR) in nasopharyngeal samples [10,11,12]. However, saliva samples have also been employed to supplement the diagnosis of these three viruses, yielding outcomes comparable to nasopharyngeal samples [13,14]. Unfortunately, conducting singleplex RT-qPCR for the three viruses can be labor-intensive and time-consuming, resulting in delayed patient management. Multiplex RT-qPCR, which targets multiple viruses in the same sample, presents an ideal solution to this issue [15]. The integration of multiplex RT-qPCR assays for SARS-CoV-2, influenza viruses, and RSV into an automated sample-to-result platform, as evaluated in other studies [16,17,18,19], markedly enhances turnaround times, facilitates concurrent screening of multiple samples, increases the reliability of results, and reduces the risk of accidental exposure for laboratory staff.

As the advancement of new laboratory diagnostic tests (LDTs) for the multiplex diagnosis of respiratory viruses progresses, it becomes crucial to incorporate them into the various molecular workflows specifically tailored for SARS-CoV-2 testing. Assessing the performance of these integrated systems will equip clinical laboratories across different regions with dependable diagnostic alternatives amidst the likely co-circulation of respiratory viruses. In this study, we scrutinized the performance of a combination of geneLEAD VIII (Precision System Science Co., Ltd., Matsudo, Japan), a fully automated sample-to-result precision instrument, and the VIASURE SARS-CoV-2, Flu & RSV Real Time PCR Detection Kit (CerTest Biotec, S.L., Zaragoza, Spain), collectively referred to as geneLEAD/VIASURE. This combination was used for the multiplex molecular diagnosis of SARS-CoV-2, IAV/IBV, and RSV. The performance of this combination was benchmarked against those of referent manual singleplex RT-qPCR assays.

## 2. Materials and Methods

### 2.1. Viruses and Preparation of Virus Stocks

This study employed an ancestral strain along with five variant strains of SARS-CoV-2, two subtype strains of endemic human IAV (H1N1 and H3N2), two strains of H5N1 subtype highly pathogenic avian influenza virus (HPAIV) isolated in 2004 and 2022 [20], two lineage strains of IBV (Yamagata and Victoria lineages), and two subtype strains of RSV (Types A and B), as outlined in Table 1. IAV was propagated by inoculating 10-day-old embryonated chicken eggs, followed by a 3-day incubation at 37 °C. The viral titer (50% tissue culture infectious dose [TCID_50_]/mL) of the harvested allantoic fluids was determined by observing the cytopathic effects in ten-fold serially diluted virus-inoculated MDCK cells cultured in a previously described virus growth medium (VGM) [21]. The MDCK cells were provided by Dr. H. Nagano (Hokkaido Institute of Public Health, Sapporo, Japan). SARS-CoV-2 was propagated by inoculating VeroE6/TMPRSS2 cells [22], which were cultured in a previously described VGM [23]. VeroE6/TMPRSS2 cells were procured from the Japanese Collection of Research Bioresources (Ibaraki, Japan). IBV was propagated by inoculating MCDK cells, which were cultured in a VGM similar to that used for IAV. RSV was propagated by inoculating HEp-2 cells procured from ATCC (Manassas, VA, USA). The VGM for RSV was composed of Dulbecco’s Modified Eagle’s Medium (Nissui Pharmaceutical Co., Ltd., Tokyo, Japan), supplemented with 1% fetal bovine serum, 2 mM L-glutamine (Fujifilm Wako Pure Chemical Industries, Ltd., Osaka, Japan), 0.15% NaHCO_3_ (Fujifilm Wako Pure Chemical Industries, Ltd., Osaka, Japan), 100 µg/mL kanamycin (Meiji Seika Pharma Co., Ltd., Tokyo, Japan), and 2 µg/mL amphotericin B (Bristol-Myers Squibb Co., New York, NY, USA). The viral titers of the harvested cell supernatant containing SARS-CoV-2, IBV, and RSV were determined by observing the cytopathic effects in ten-fold serially diluted virus-inoculated VeroE6/TMPRSS2, MDCK, and Hep-2 cells, respectively. These cell supernatants containing SARS-CoV-2, IBV, and RSV, along with the allantoic fluids containing IAV, were aliquoted and stored at −80 °C for use as virus stock solutions.

### 2.2. Viral Gene Detection Using geneLEAD/VIASURE

The VIASURE SARS-CoV-2, Flu & RSV Real Time PCR Detection Kit comprises specific primers and fluorescence-labeled probes targeting the *nucleocapsid (N)* gene of SARS-CoV-2, the *matrix 1 (M1)* gene of IAV/IBV, and the *N* gene of RSV. It also includes a housekeeping gene (human *RNase P* gene) as an endogenous internal control (IC). As the VIASURE SARS-CoV-2, Flu & RSV Real Time PCR Detection Kit is proprietary, the primer/probe sequence details were not provided. The experiments in this study were conducted using the geneLEAD VIII system, which allows for minimal hands-on processing time and simultaneous analysis of 8 samples in approximately 2 h. The experiments were performed using virus-spiked saliva samples (Normal Saliva, Pooled Human Donors, Lee Biosolutions Inc., Maryland Heights, MO, USA) or pooled nasal swab solutions obtained from two volunteers in our laboratory. Collection of nasal swabs and their use in the experiments took place in April 2025 after receiving ethical approval from the Institutional Ethics Committee of Obihiro University of Agriculture and Veterinary Medicine. To collect the nasal swabs, sterile cotton swabs were inserted about 2 cm into the nasal cavity, rotated for approximately five times, and left for 5 min. Samples were collected from both nostrils without changing the swab. Subsequently, the swab from each volunteer was soaked in 1 mL phosphate-buffered saline (PBS) to prepare the nasal swab solutions. These swab solutions were then mixed and confirmed to be negative for SARS-CoV-2, IAV, IBV, and RSV using the geneLEAD/VIASURE. The virus-spiked saliva sample or the nasal swab solution was mixed with Prep Buffer A/PBS at a 7:13 ratio. Prep Buffer A/PBS was prepared beforehand by mixing Prep Buffer A (Precision System Science Co., Ltd.), a commercial virus lysis/transport buffer, with PBS at a 5:8 ratio, as previously described [24]. After incubation at 25 °C for at least 10 min for virus inactivation, all the following procedures were performed automatically using geneLEAD/VIASURE. In brief, 50 µL of RNA was extracted from 49 µL of virus-spiked saliva or the pooled nasal swab solutions using the geneLEAD VIII system with MagDEA Dx SV (Precision System Science Co., Ltd.), as previously described [24]. A one-step RT-qPCR was then performed using 5 µL of the extracted RNA samples and 15 µL of the VIASURE Kit’s reaction mix. The one-step RT-qPCR conditions included reverse transcription at 45 °C for 15 min, initial denaturation at 95 °C for 2 min, and 45 cycles of 95 °C for 10 s (denaturation) and 63 °C for 50 s (annealing/extension). Fluorogenic data from successful amplification for SARS-CoV-2, IAV/IBV, and RSV gene targets were detected in the FAM, ROX, and Cy5 channels, respectively. The IC gene was detected in the HEX channel. Cycle threshold (Ct) values of ≤40 for the four virus targets and either a Ct value of ≤40 or no signal for the IC (since a high copy number of the target can cause preferential amplification of the target-specific nucleic acids) were considered valid positive results. Conversely, tests were deemed valid negative results only if they had a Ct value of >40 or no signal for target viral genes, and the IC exhibited an amplification signal with a Ct value of ≤35. These criteria are based on the manufacturer’s instructions provided with the VIASURE kits (Precision System Science Co., Ltd., Chiba, Japan).

### 2.3. Reference Assays

Singleplex one-step RT-qPCR assays for the SARS-CoV-2 *N* gene, IAV *M1* gene, IBV *nonstructural (NS)* gene, and RSV *M* gene were utilized in this study, each employing primers/probe, as detailed in Appendix A. These assays, recommended by the National Institute of Infectious Diseases (NIID) in Japan [25,26,27], served as the reference assays and are referred to as the manual assays in this study. For these manual assays, virus-spiked saliva samples were mixed with Prep Buffer A/PBS and incubated, as described in Section 2.2. Then, 50 µL of RNA was extracted from 49 µL of the saliva samples using the QIAamp Viral RNA Kit (QIAGEN N.V., Venlo, The Netherlands), following the manufacturer’s guidelines. A total of 5 µL of the extracted RNA was then subjected to one-step RT-qPCR in the LightCycler^®^ 96 instrument (Roche Diagnostics, Basel, Switzerland) using the QuantiTect Probe RT-PCR Kit (QIAGEN), as previously described [24]. The PCR conditions for each virus target are summarized in Appendix A. As a control, RNA extracted from virus-free saliva was included in each one-step RT-qPCR. The cutoff values for the manual assays were determined using the method-based Ct cutpoint [28]. Briefly, saliva samples spiked with serial dilutions of the virus stock and virus-free samples were analyzed by the manual assays. Then, the amplification curves of both samples were compared, and the cutoff values were determined based on the Ct values corresponding to a reliable detection threshold. These cutoff Ct values for SARS-CoV-2, IAV, IBV, and RSV were set at 40.0, 38.5, 38.5, and 37.5, respectively.

### 2.4. Determination of the Analytical Reactivity and the Viral Titer Detection Limit of geneLEAD/VIASURE

The analytical reactivity of geneLEAD/VIASURE was assessed by testing a selection of representative strains of SARS-CoV-2, IAV, IBV, and RSV. Stock solutions containing viruses with varying viral titers were diluted 100-fold in PBS and subsequently spiked in saliva samples or pooled nasal swab solutions in a 1:9 ratio. These virus-spiked samples were then mixed with Prep Buffer A/PBS, as described in Section 2.2, and 49 µL of these virus-spiked samples was forwarded for analysis using either geneLEAD/VIASURE or the manual assays to confirm the specific detection of individual virus genes.

To analyze the viral titer detection limit of geneLEAD/VIASURE, we used the SARS-CoV-2 ancestral strain, H1N1-subtype IAV, Yamagata-lineage IBV, and Type A RSV. The stock solutions of SARS-CoV-2 and IBV were serially diluted in PBS from 10^4^–10^0^ TCID_50_/mL, IAV from 10^4^–10^−1^ TCID_50_/mL, and RSV from 10^6.25^–10^0.25^ TCID_50_/mL. The diluted solutions of SARS-CoV-2, IAV, and IBV were spiked into saliva at a ratio of 1:9, while the RSV solution was spiked at a ratio of 1:1. The resulting viral titers in the virus-spiked saliva samples were 10^3^–10^−1^ TCID_50_/mL for SARS-CoV-2 and IBV, 10^3^–10^−2^ TCID_50_/mL for IAV, and approximately 10^6^–10^0^ TCID_50_/mL for RSV. The virus-spiked saliva samples were then analyzed using either geneLEAD/VIASURE or the manual assays, and the Ct values were compared.

### 2.5. Analysis of Single-, Double-, and Triple-Infection Models

The studies utilized the SARS-CoV-2 ancestral strain, H1N1-subtype IAV, Yamagata-lineage IBV, and type A RSV. In the single-infection model studies, either stock or PBS-diluted virus solutions containing 10^4^ TCID_50_/mL for SARS-CoV-2, IAV, and IBV and 10^6.25^ TCID_50_/mL for RSV were used for the high-viral-titer group. For the low-viral-titer group, virus solutions containing 10^2^ TCID_50_/mL for SARS-CoV-2, IAV, and IBV and 10^4.25^ TCID_50_/mL for RSV were used. These were spiked into saliva in the ratios outlined in the previous section. For the double and triple infections involving RSV, we diluted SARS-CoV-2, IAV, and IBV solutions in saliva instead of PBS to maintain a nearly constant saliva concentration across the single- and co-infection models. Several panels of saliva samples were prepared, each containing spiked viruses of either uniformly high (10^3^ TCID_50_/mL for SARS-CoV-2, IAV, and IBV and 10^6^ TCID_50_/mL for RSV) or uniformly low (10^1^ TCID_50_/mL for SARS-CoV-2, IAV, and IBV and 10^4^ TCID_50_/mL for RSV) viral titer. Additionally, panels of saliva samples spiked with varying viral titers (a combination of high and low viral titers) were prepared. Notably, however, as geneLEAD/VIASURE detects both IAV and IBV in the same channel and cannot differentiate between them, we did not test coinfection models containing these two viruses. The capacity of geneLEAD/VIASURE to concurrently detect SARS-CoV-2, IAV/IBV, and RSV in the same sample matrix was evaluated by comparing Ct values of saliva samples spiked with individual viruses to those spiked with the corresponding double or triple viruses.

### 2.6. Correlation Between geneLEAD/VIASURE and the Manual Assays

In addition to the virus-spiked saliva samples utilized in the analytical reactivity and viral titer detection limit studies, all sample panels examined in the single-, double-, or triple-infection models using geneLEAD/VIASURE were also subjected to analysis using the manual assays. The Ct values derived from both assays were employed to evaluate their correlation, as demonstrated in the subsequent section.

### 2.7. Data Analysis

The Ct values obtained from both geneLEAD/VIASURE and the manual assays (*n* = 2–4) were input into Microsoft Excel 2019 software (Microsoft Corporation, Redmond, WA, USA), where the mean and standard deviation were computed. Simple linear regression was applied to correlate the Ct values acquired by geneLEAD/VIASURE with those obtained by the manual assays. The Pearson’s coefficient of determination (R^2^) was utilized to determine the concordance between the two methods. All graphs were generated using GraphPad Prism version 10.2.2 (GraphPad Software Inc., San Diego, CA, USA).

## 3. Results

### 3.1. Analytical Reactivity and the Viral Titer Detection Limit of geneLEAD/VIASURE

The geneLEAD/VIASURE successfully detected the genes of all the tested SARS-CoV-2 strains, two endemic subtype (H1N1 and H3N2) strains of human IAV, two lineage (Yamagata and Victoria) strains of IBV, and two subtype (Type A and Type B) strains of RSV in their respective detection channels in virus-spiked saliva or pooled nasal swab samples. Additionally, two strains of H5N1-subtype HPAIV, isolated nearly two decades apart, were specifically detected in the respective ROX channel. Notably, using the virus-spiked saliva samples, geneLEAD/VIASURE yielded lower Ct values compared to the manual assays for all the viral strains tested (Table 2). On the other hand, the manual assay results for the virus-spiked saliva samples were comparable with the geneLEAD/VIASURE results for the virus-spiked nasal swab solutions for many tested viruses. The exceptions were the IAV and IBV Yamagata lineage strains, whose Ct values were much higher in the manual assay (Table 2). Notably, in the virus-spiked saliva samples and nasal swab solutions, geneLEAD/VIASURE showed specific amplification of the spiked viruses but not others.

In the viral titer detection limit study, geneLEAD/VIASURE did not exhibit viral gene amplification for the virus-free saliva-negative control. Conversely, the manual assays demonstrated nonspecific amplification for all targets, with the exception of SARS-CoV-2 (Figure 1). For geneLEAD/VIASURE, the viral titer detection limits for SARS-CoV-2, IAV, IBV, and RSV were 10^0^, ≤10^−2^, 10^0^, and 10^2^ TCID_50_/mL, respectively. In the manual assays, the lowest viral titers that resulted in a Ct value above the cutoff for SARS-CoV-2, IAV, IBV, and RSV were considered the virus titer detection limits, which were ≤10^−1^, ≤10^0^, ≤10^0^, and ≤10^4^ TCID_50_/mL, respectively (Figure 1).

### 3.2. Simultaneous Detection of Single, Double, and Triple Viral Genes

Saliva samples spiked with either one or a combination of two or three different viruses with constant viral titers (all the titers were uniformly high or low), representing single-, double-, or triple-infection models, were evaluated. The Ct values of SARS-CoV-2, IAV, IBV, and RSV obtained using geneLEAD/VIASURE in the single-infection model were consistent with those in both the double- and triple-infection models under both high and low-viral-titer conditions (Table 3). Additionally, we evaluated various combinations of high and low viral titers in the triple-infection models to rigorously test the multiplexing capability of geneLEAD/VIASURE. While the SARS-CoV-2 and IAV/IBV Ct values remained unaffected in all the combinations, our results showed that the RSV Ct value of the samples containing a low titer of RSV mixed with high titers of either SARS-CoV-2 or IAV/IBV (Table 4) was slightly higher compared to that observed in the low-titer single RSV infection model (Table 3).

### 3.3. Agreement Between geneLEAD/VIASURE and the Manual Assays

The Ct values from 62 SARS-CoV-2-spiked, 44 IAV-spiked, 41 IBV-spiked, and 56 RSV-spiked saliva samples, which were concurrently tested using geneLEAD/VIASURE and manual assays, were used for the correlation analyses. Overall, a commendable linear correlation was observed between these two assays, with SARS-CoV-2, IAV, IBV, and RSV exhibiting Pearson’s R^2^ values of 0.90, 0.85, 0.92, and 0.95, respectively (Figure 2).

## 4. Discussion

We hereby assessed the performance of the geneLEAD VIII platform, whose open modus, an intrinsic functionality, permits integration with LDTs [29]. The geneLEAD VIII system has been previously utilized for SARS-CoV-2 detection in various experimental configurations [24,29,30]. However, to our knowledge, its efficacy in multiplex diagnostics remains unexplored. Similarly, while certain VIASURE kit series have undergone evaluation for the multiplex diagnosis of IAV, IBV, and RSV [31] or the multiplex detection of two different SARS-CoV-2 genes and an IC [32], no research exists that reports the multiplex performance of the VIASURE SARS-CoV-2, Flu & RSV Real Time PCR Detection Kit, capable of detecting SARS-CoV-2, IAV/IBV, and RSV in a sample. Consequently, this study was undertaken to evaluate the performance of geneLEAD/VIASURE in the multiplex diagnosis of SARS-CoV-2, IAV/IBV, and RSV.

In this study, geneLEAD/VIASURE was used to specifically detect all the examined SARS-CoV-2 strains, belonging to multiple lineages, endemic human IAVs of subtypes H1N1 and H3N2, two lineage strains of IBV, and two subtype strains of RSV in their respective channels using virus-spiked saliva samples and pooled nasal swab solutions (Table 2). Additionally, it identified two H5N1-subtype HPAIV strains isolated nearly two decades apart, which have been consistently linked to sporadic human cases [33]. Compared to manual assays, the geneLEAD/VIASURE showed an inferior viral titer detection limit for SARS-CoV-2, an equivalent viral titer detection limit for IBV, and a superior viral titer detection limit for IAV and RSV. Importantly, geneLEAD/VIASURE did not yield nonspecific Ct values in virus-free samples, and the Ct values for all the test viruses spiked in saliva were slightly lower (albeit to varying degrees between variants, subtypes, and lineages) than those of the manual assays (Figure 1, Table 2). These findings suggest that geneLEAD/VIASURE exhibits superior performance in detecting target genes in saliva samples with low viral titers. Nevertheless, regarding the limitation of this study, information on the VIASURE kit’s primer and probe sequences was not provided due to proprietary issues, which makes it difficult to comprehensively compare the performance of geneLEAD/VIASURE and the manual assays.

The Ct values of the virus-spiked nasal swab solutions were slightly higher than those of the virus-spiked saliva samples in the geneLEAD/VIASURE analysis, but comparable with the Ct values of the saliva samples analyzed using the manual assays, except for the IAV and IBV Yamagata lineage strains. The Ct values of these five strains in the nasal swab solutions obtained with geneLEAD/VIASURE were much lower than those of the saliva samples analyzed using the manual assay. Although we were unable to investigate this difference experimentally, we hypothesize that this might have been due to the differences in the compositions of the samples, such as RNases or other inhibitors that would affect RT-qPCR.

During the performance of a multiplex PCR, certain targets may be amplified more efficiently than others due to factors such as the PCR reagents, the properties of the target genes, and variations in the thermal profiles of a thermocycler [34]. Furthermore, the precision of the instrument and its ability to distinguish fluorescent signals from different targets in multiplex RT-qPCR is a critical performance characteristic that warrants comprehensive investigation [35]. Given that geneLEAD/VIASURE is designed for use in situations in which co-infection with SARS-CoV-2, IAV, IBV, and/or RSV is possible, a significant aspect of this study was the assessment of its efficacy in detecting double- and/or triple infections compared to a single infection. Our analysis revealed that the Ct values for the four viruses in the single-infection models were highly consistent with those in the double- and triple-infection models when the spiked viruses had constant (all high or low) titers (Table 3), underscoring the reliability of geneLEAD/VIASURE. Nonetheless, as the titers of each virus in clinical specimens obtained from co-infected patients are different, we aimed to rigorously assess the multiplexing capability of geneLEAD/VIASURE by spiking viruses with varying titers in the triple-infection models. Our findings revealed that the RSV Ct values of the samples spiked with low titers of RSV and higher titers of SARS-CoV-2, IAV, or IBV were slightly higher than those obtained in the RSV single infection model. This might have been due to the preferential amplification of the target genes of viruses with high viral titers, as shown in a recently developed multiplex RT-qPCR assay for the simultaneous detection of H5-, H7-, and H9-subtype avian influenza viruses [36]. As highlighted by An et al. [36], reducing the primer concentration of the preferentially amplified targets or using other reagents more suitable for multiplexing can improve this situation.

Additionally, optimal concordance for SARS-CoV-2, IBV, and RSV (R^2^ ≥ 90%) was observed between the Ct values from the geneLEAD/VIASURE analysis and those obtained using the manual assays, while suboptimal concordance for IAV was noted (R^2^ = 85%) (Figure 2). Intriguingly, saliva samples spiked with low viral titers of IAV exhibited a variable range of Ct values following analysis by the manual assay, which differed from the automated detection system. This observation prompts questions about the efficiency of the manual assay in amplifying low-viral-titer IAV samples.

In many clinical diagnostic laboratories, sufficient numbers of clinical samples containing multiple viruses are currently not readily available. As a result, we were unable to obtain clinical samples collected from SARS-CoV-2, IAV, IBV, and RSV multiple-infected patients. This limits our study, as evaluating the system solely using virus-spiked samples obtained from healthy individuals remains unsatisfactory. Clinical specimens are more complex and may contain inhibitors or interfering substances that could affect assay performance. Nevertheless, the trends observed in our study, in which viruses with different titers were spiked into saliva and nasal swab samples, may provide some similarities with results from clinical specimens. Our results may demonstrate the potential reliability of geneLEAD/VIASURE in diagnosing co-infections caused by common respiratory viruses amid their likely co-circulation. Future studies using clinical samples from co-infected patients should be conducted to confirm the clinical validity of geneLEAD/VIASURE, which will also help to determine its performance metrics, such as specificity, sensitivity, and clinical accuracy.

In conclusion, geneLEAD/VIASURE has demonstrated itself to be a highly specific, sensitive, reliable, and reproducible platform for the detection of SARS-CoV-2, IAV, IBV, and RSV, whether as single infections or co-infections, as anticipated in clinical settings.

## Figures and Tables

**Figure 1 diagnostics-15-01791-f001:**
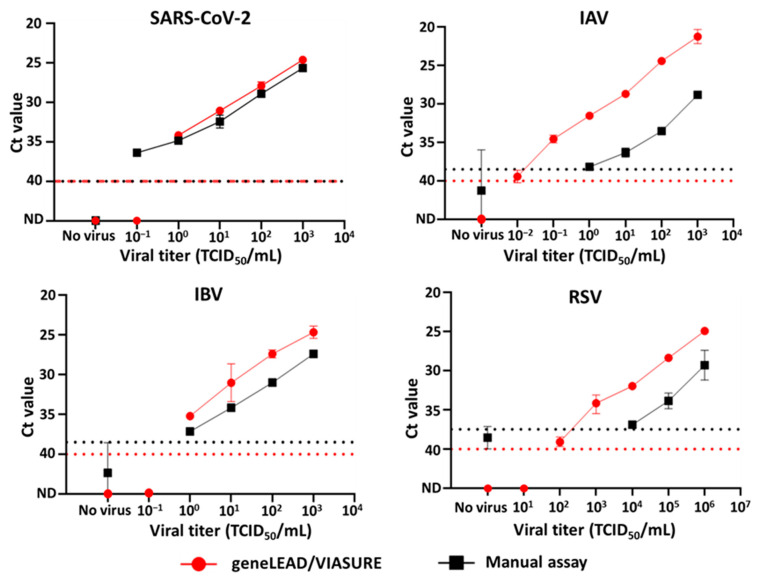
Comparison of the viral titer detection limits of SARS-CoV-2 (*n* = 2), IAV (*n* = 2), IBV (*n* = 2), and RSV (*n* = 2–4) between analyses conducted using geneLEAD/VIASURE (●) and the manual assays (■). The red and black dotted lines denote the cutoff Ct values for the automated system and the manual assays, respectively.

**Figure 2 diagnostics-15-01791-f002:**
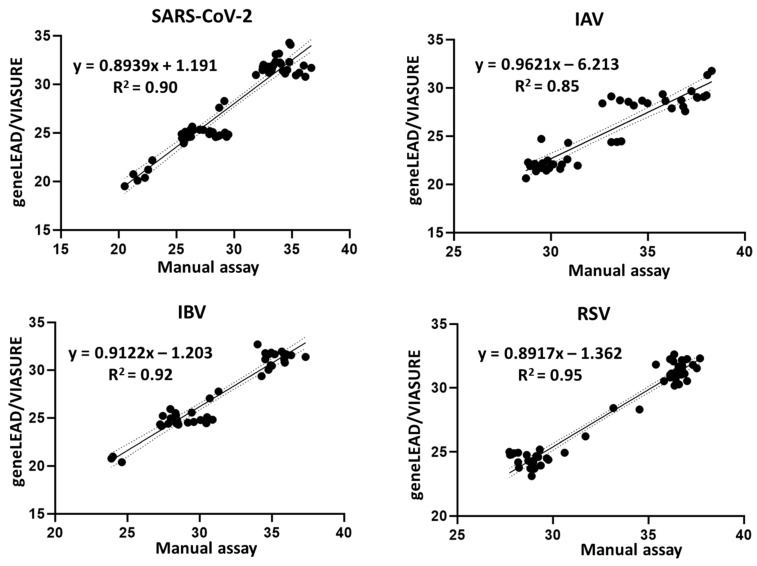
Simple linear regression plots of Ct values for the detection of SARS-CoV-2 (*n* = 62), IAV (*n* = 44), IBV (*n* = 41), and RSV (*n* = 56), obtained using geneLEAD/VIASURE versus the manual assays. Each plot displays the equation for the best-fit line and Pearson’s coefficient of determination (R^2^). The dotted lines indicate the 95% confidence bands of the best-fit line.

**Table 1 diagnostics-15-01791-t001:** SARS-CoV-2, IAV, IBV, and RSV strains tested in this study with geneLEAD/VIASURE.

Virus	Variant, Subtype, or Lineage	Strain Name	GISAID ID/NCBI Accession No.	Source
SARS-CoV-2	Ancestral (A ^a^)	2019-nCoV/Japan/TY/WK-521/2020	EPI_ISL_408667 ^c^	NIID
	Alpha (B.1.1.7 ^a^)	hCoV-19/Japan/QHN001/2020	EPI_ISL_804007 ^c^	NIID
	Delta (B.1.716.2 ^a^)	hCoV-19/Japan/TY11-927/2021	EPI_ISL_2158617 ^c^	NIID
	Gamma (P.1 ^a^)	hCoV-19/Japan/TY7-501/2021	EPI_ISL_833366 ^c^	NIID
	Beta (B.1.351 ^a^)	hCoV-19/Japan/TY8-612/2021	EPI_ISL_1123289 ^c^	NIID
	Omicron (BA.5 ^a^)	hCoV-19/Japan/TY41-702/2022	EPI_ISL_13241867 ^c^	NIID
IAV	H1N1	A/Narita/1/2009	EPI_ISL_30176 ^c^	NIID
	H3N2	A/Hokkaido/19/1998	Not available	HIPH
	H5N1 (HPAIV, 2022 ^b^)	A/white-tailed eagle/Japan/OU-1/2022	LC775579 ^d^	Our lab [20]
	H5N1 (HPAIV, 2004 ^b^)	A/chicken/Yamaguchi/7/04	AB166862 ^d^	NIAH
IBV	Yamagata-lineage	B/Hokkaido/25/2018	Not available	HIPH
	Victoria-lineage	B/Hokkaido/1/2019	EPI_ISL_363734 ^c^	HIPH
RSV	Type A	RSV/A/NIID/2370/14	LC474558 ^e^	NIID
	Type B	RSV/B/NIID/2472/14	LC474559 ^e^	NIID

^a^ PANGO lineage; ^b^ Isolated year; ^c^ GISAID ID; ^d^ GenBank accession no. for the hemagglutinin gene; ^e^ GenBank accession no. for nearly complete genome; NIID: National Institute of Infectious Diseases (Tokyo, Japan); NIAH: National Institute of Animal Health (Tsukuba, Japan); HIPH: Hokkaido Institute of Public Health (Sapporo, Japan).

**Table 2 diagnostics-15-01791-t002:** Specific detection of SARS-CoV-2, IAV, IBV, and RSV strains in saliva or nasal swab solutions via geneLEAD/VIASURE and manual assays (*n* = 2).

Virus (*n* = 2)	Manual Assays (Ct ± SD)	geneLEAD/VIASURE (Ct ± SD)	
	Virus-Spiked Saliva	Virus-Spiked Saliva	Virus-Spiked Nasal Swab Solutions
SARS-CoV-2 (Ancestral)	22.92 ± 0.18	22.18 ± 0.34	23.85 ± 0.24
SARS-CoV-2 (Alpha)	21.65 ± 0.28	20.09 ± 0.00	23.32 ± 0.04
SARS-CoV-2 (Delta)	22.29 ± 0.54	20.38 ± 0.14	22.29 ± 0.01
SARS-CoV-2 (Gamma)	21.28 ± 0.24	20.74 ± 0.52	21.67 ± 0.41
SARS-CoV-2 (Beta)	22.56 ± 0.66	21.21 ± 0.35	22.15 ± 0.35
SARS-CoV-2 (Omicron)	20.53 ± 0.49	19.50 ± 0.21	21.25 ± 0.47
IAV (H1N1)	28.83 ± 0.16	22.27 ± 0.42	23.10 ± 0.64
IAV (H3N2)	30.49 ± 0.01	21.61 ± 1.08	20.97 ± 0.09
AIV (H5N1, 2022 ^a^)	30.90 ± 0.09	24.30 ± 0.06	24.61 ± 0.25
AIV (H5N1, 2004 ^a^)	33.12 ± 0.04	24.37 ± 0.01	25.36 ± 0.21
IBV (Yamagata lineage)	24.63 ± 0.21	20.40 ± 0.03	20.54 ± 0.17
IBV (Victoria lineage)	28.49 ± 0.24	27.99 ± 0.42	29.41 ± 0.39
RSV (Type A)	31.71 ± 0.16	26.21 ± 0.45	28.37 ± 0.43
RSV (Type B)	35.40 ± 0.21	31.82 ± 0.67	34.24 ± 0.30

^a^ Isolated year.

**Table 3 diagnostics-15-01791-t003:** Concurrent detection of single-, double-, and triple-infection models with constant viral titers using geneLEAD/VIASURE (*n* = 4).

Viruses Spiked in Saliva Samples and Their Viral Titers (TCID_50_/mL)				Ct Value (±SD) Obtained Using geneLEAD/VIASURE			
SARS-CoV-2	IAV	IBV	RSV	SARS-CoV-2 ^a^ (FAM)	IAV ^a^ (ROX)	IBV ^a^ (ROX)	RSV ^a^ (Cy5)
10^3^ (high)	-	-	-	24.90 ± 0.25	n/d	n/d	n/d
-	10^3^ (high)	-	-	n/d ^b^	22.66 ± 1.38	n/d	n/d
-	-	10^3^ (high)	-	n/d	n/d	24.90 ± 0.74	n/d
-	-	-	10^6^ (high)	n/d	n/d	n/d	24.67 ± 0.26
10^3^ (high)	10^3^ (high)	-	-	25.03 ± 0.33	21.91 ± 0.46	n/d	n/d
10^3^ (high)	-	10^3^ (high)	-	24.84 ± 0.27	n/d	24.76 ± 0.52	n/d
10^3^ (high)	-	-	10^6^ (high)	25.02 ± 0.30	n/d	n/d	24.56 ± 0.40
-	10^3^ (high)	-	10^6^ (high)	n/d	22.17 ± 0.26	n/d	24.12 ± 0.15
-	-	10^3^ (high)	10^6^ (high)	n/d	n/d	24.96 ± 0.52	24.81 ± 0.26
10^3^ (high)	10^3^ (high)	-	10^6^ (high)	24.47 ± 0.40	21.78 ± 0.32	n/d	23.57 ± 0.30
10^3^ (high)	-	10^3^ (high)	10^6^ (high)	25.27 ± 0.26	n/d	25.27 ± 0.27	23.83 ± 0.25
10^1^ (low)	-	-	-	31.88 ± 1.02	n/d	n/d	n/d
-	10^1^ (low)	-	-	n/d	29.23 ± 0.31	n/d	n/d
-	-	10^1^ (low)	-	n/d	n/d	30.60 ± 0.45	n/d
-	-	-	10^4^ (low)	n/d	n/d	n/d	32.03 ± 0.65
10^1^ (low)	10^1^ (low)	-	-	31.68 ± 0.15	28.55 ± 0.40	n/d	n/d
10^1^ (low)	-	10^1^ (low)	-	31.24 ± 0.39	n/d	31.66 ± 0.22	n/d
10^1^ (low)	-	-	10^4^ (low)	31.95 ± 0.35	n/d	n/d	31.55 ± 0.53
-	10^1^ (low)	-	10^4^ (low)	n/d	28.14 ± 0.64	n/d	31.31 ± 0.53
-	-	10^1^ (low)	10^4^ (low)	n/d	n/d	31.30 ± 0.60	31.74 ± 0.27
10^1^ (low)	10^1^ (low)	-	10^4^ (low)	31.74 ± 0.31	28.78 ± 0.39	n/d	30.44 ± 0.22
10^1^ (low)	-	10^1^ (low)	10^4^ (low)	32.21 ± 0.65	n/d	31.55 ± 0.29	30.65 ± 0.32

^a^ Viruses of which genes were targeted in the analyses using geneLEAD/VIASURE. ^b^ Not detected (Ct ≥ 40).

**Table 4 diagnostics-15-01791-t004:** Concurrent detection of triple-infection models containing various combinations of viral titers using geneLEAD/VIASURE (*n* = 4).

Viruses Spiked in Saliva Samples and Their Viral Titers (TCID_50_/mL)				Ct Value (±SD) Obtained Using geneLEAD/VIASURE			
SARS-CoV-2	IAV	IBV	RSV	SARS-CoV-2 ^a^ (FAM)	IAV ^a^ (ROX)	IBV ^a^ (ROX)	RSV ^a^ (Cy5)
10^3^ (high)	10^1^ (low)	-	10^4^ (low)	24.89 ± 0.38	30.17 ± 0.46	n/d	30.43 ± 1.53
10^3^ (high)	-	10^1^ (low)	10^4^ (low)	25.17 ± 0.38	n/d ^b^	32.16 ± 1.31	33.28 ± 0.27
10^3^ (high)	10^3^ (high)	**-**	10^4^ (low)	24.82 ± 0.39	23.50 ± 0.42	n/d	35.46 ± 1.21
10^3^ (high)	**-**	10^3^ (high)	10^4^ (low)	24.91 ± 0.28	n/d	23.88 ± 0.26	36.09 ± 2.03
10^1^ (low)	10^3^ (high)	**-**	10^4^ (low)	32.18 ± 0.85	22.4 ± 1.29	n/d	32.84 ± 0.03
10^1^ (low)	**-**	10^3^ (high)	10^4^ (low)	33.06 ± 0.38	n/d	23.68 ± 0.94	34.65 ± 0.30
10^1^ (low)	10^1^ (low)	**-**	10^6^ (high)	31.25 ± 0.15	27.72 ± 0.22	n/d	23.77 ± 0.14
10^1^ (low)	**-**	10^1^ (low)	10^6^ (high)	31.36 ± 0.21	n/d	32.65 ± 0.76	23.63 ± 0.43
10^1^ (low)	10^3^ (high)	**-**	10^6^ (high)	31.44 ± 0.39	24.02 ± 0.42	n/d	23.88 ± 0.16
10^1^ (low)	**-**	10^3^ (high)	10^6^ (high)	32.51 ± 0.85	n/d	24.72 ± 1.68	24.73 ± 0.67

^a^ Viruses of which genes were targeted in the analyses using geneLEAD/VIASURE. ^b^ Not detected (Ct > 40). The Ct values highlighted in blue are slightly higher (Ct value ≥ 1.5) compared with those of the equivalent titers in single infections (Table 3).

## Data Availability

Data will be made available on request.

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
