# Peer review of "Performance Evaluation of a Fully Automated Molecular Diagnostic System for Multiplex Detection of SARS-CoV-2, Influenza A/B Viruses, and Respiratory Syncytial Virus"

_diagnostics, 2025, doi:10.3390/diagnostics15141791_

Round 1
Reviewer 1 Report (Previous Reviewer 3)
Comments and Suggestions for Authors
The work has improved significantly. I agree with the authors when they say that "While it is true that many automated sample-to-result platforms are available for the multiplexed diagnosis of viruses, they are not readily accessible for routine diagnosis in many clinical laboratories." For this reason, I have changed my opinion about this article and believe that it is indeed helpful to the clinical-scientific community.
Author Response
Reviewer 1: Comments and suggestions for authors
Comment 1: The work has improved significantly. I agree with the authors when they say that "While it is true that many automated sample-to-result platforms are available for the multiplexed diagnosis of viruses, they are not readily accessible for routine diagnosis in many clinical laboratories." For this reason, I have changed my opinion about this article and believe that it is indeed helpful to the clinical-scientific community.
Response 1: We thank the reviewer for the positive feedback.
Reviewer 2 Report (Previous Reviewer 2)
Comments and Suggestions for Authors
In the mansucript with No. diagnostics-3688028, the authors evaluated the performance of a fully automated sample-to-result system in combination with a real-time PCR detection kit for SARS-CoV-2, Flu, and RSV. They further compared this automated system with a manual assay based on primer/probe sequences recommended by the National Institute of Infectious Diseases (NIID) in Japan. Unfortunately, I do not believe the manuscript is currently suitable for publication in Diagnostics, due to the following concerns:
1. Lack of detailed primer/probe information: There was no information provided regarding the primer and probe sequences used in the automated system for the three viruses. Since the system may include multiple primer/probe sets for a single virus, the absence of this information makes it difficult to meaningfully compare the performance of the automated system with the real time kit with that of the manual assay.
2. Lack of clinical sample evaluation: A major limitation of the study is the absence of clinical samples. Clinical specimens are more complex and often contain potential inhibitors or interfering substances, which may affect assay performance. Evaluating the system solely using virus-spiked samples from healthy individuals limits the study.
Author Response
We are greatly indebted to the reviewers for their thorough review of our manuscript and for providing us with insightful comments and suggestions. Subsequently, we have attentively revised the manuscript and highlighted all the changes in red. In addition, our point-to-point responses to the reviewers’ comments are provided below:
Reviewer 2: Comments and suggestions for authors
In the mansucript with No. diagnostics-3688028, the authors evaluated the performance of a fully automated sample-to-result system in combination with a real-time PCR detection kit for SARS-CoV-2, Flu, and RSV. They further compared this automated system with a manual assay based on primer/probe sequences recommended by the National Institute of Infectious Diseases (NIID) in Japan. Unfortunately, I do not believe the manuscript is currently suitable for publication in Diagnostics, due to the following concerns:
Comment 1: Lack of detailed primer/probe information: There was no information provided regarding the primer and probe sequences used in the automated system for the three viruses. Since the system may include multiple primer/probe sets for a single virus, the absence of this information makes it difficult to meaningfully compare the performance of the automated system with the real time kit with that of the manual assay.
Response 1: Thank you for the concern. Unfortunately, the VIASURE SARS-CoV-2, Flu & RSV Real Time PCR Detection Kit (CerTest Biotec, S.L., Zaragoza, Spain) is proprietary; thus, the primer/probe sequences were not provided. These details are now included in the manuscript (lines 117–119). The only provided information was on the targeted gene regions for each virus (lines 114–117) and the fluorescent dyes used to label the different probes (lines 141–144).
Comment 2: Lack of clinical sample evaluation: A major limitation of the study is the absence of clinical samples. Clinical specimens are more complex and often contain potential inhibitors or interfering substances, which may affect assay performance. Evaluating the system solely using virus-spiked samples from healthy individuals limits the study.
Response 2: We agree with the reviewer that the use of virus-spiked samples was a major limitation, as acknowledged in our manuscript (lines 359–362). In many clinical diagnostic laboratories, sufficient clinical samples containing multiple viruses are not readily available. Thus, we were unable to obtain such samples and used virus-spiked ones instead. Although our data were obtained using saliva and nasal swab samples from healthy individuals, we believe the trends observed still offer some reflection of results from clinical specimens. We believe our findings provide some insight into the potential of geneLEAD/VIASURE for multiplex molecular diagnosis of viral infectious diseases. Nevertheless, we agree with the necessity of future studies using clinical samples to confirm the validity of our automated system (lines 366–369).
Reviewer 3 Report (Previous Reviewer 1)
Comments and Suggestions for Authors
The manuscript focused on the comprehensive analytical assessment of the geneLEAD VIII system coupled with the VIASURE multiplex RT-qPCR kit (geneLEAD/VIASURE) for detecting SARS-CoV-2, influenza A/B, and RSV. The work meets a clinically important need for fast, automated multiplex diagnostics among cocirculating respiratory pathogens. Although the technical validation is mostly strong, major changes before publication are required based on important restrictions on clinical applicability and interpretation of results:
- The use of virus-spiked saliva/nasal swab solutions in clinical testing is flawed due to the complex matrices, host factors, and potential co-infecting agents involved.
- The primer probe has been taken for the manual assay to compare the results what is the LOD/LOQ of the those primer probes
- Reconcile detection limit statements throughout (Abstract, Results 3.1, Discussion). Acknowledge that the automated system has equivalent or reduced analytical sensitivity for SARS-CoV-2 and RSV compared to manual methods. Avoid overstating superiority.
- Period of the study dhould be mentioned in the material methods
- Results of the negative samples need to show the specificity of the developed methods. What is the sensitivity and specificity of the developed methods? They need to be mentioned and also compared with the manual method.
- How the quantitation of the samples that were spiked was quantified for the specific virus needs to be described.
Author Response
We are greatly indebted to the reviewers for their thorough review of our manuscript and for providing us with insightful comments and suggestions. Subsequently, we have attentively revised the manuscript and highlighted all the changes in red. In addition, our point-to-point responses to the reviewers’ comments are provided below:
Reviewer 3: Comments and suggestions for authors
The manuscript focused on the comprehensive analytical assessment of the geneLEAD VIII system coupled with the VIASURE multiplex RT-qPCR kit (geneLEAD/VIASURE) for detecting SARS-CoV-2, influenza A/B, and RSV. The work meets a clinically important need for fast, automated multiplex diagnostics among cocirculating respiratory pathogens. Although the technical validation is mostly strong, major changes before publication are required based on important restrictions on clinical applicability and interpretation of results:
Comment 1: The use of virus-spiked saliva/nasal swab solutions in clinical testing is flawed due to the complex matrices, host factors, and potential co-infecting agents involved.
Response 1: We agree with the reviewer’s comment. Please see Response 2 for reviewer 2.
Comment 2: The primer probe has been taken for the manual assay to compare the results what is the LOD/LOQ of those primer probes
Response 2: Although the limit of quantification (LOQ) of the primers used in the manual assay is not specified in the NIID manuals (references 25–27), one objective of this study was to determine the viral titer detection limit of the manual assay. This was found to be ≤10⁻¹, ≤10⁰, ≤10⁰, and ≤10⁴ TCID₅₀/mL for SARS-CoV-2, IAV, IBV, and RSV, respectively (lines 243–247).
Comment 3: Reconcile detection limit statements throughout (Abstract, Results 3.1, Discussion). Acknowledge that the automated system has equivalent or reduced analytical sensitivity for SARS-CoV-2 and RSV compared to manual methods. Avoid overstating superiority.
Response 3: Thank you for highlighting this point. We have reconciled the detection limit statements in the Abstract (lines 31–33) and Results (lines 243–247). We have clearly stated that the manual system had superior viral titer detection limit for SARS-CoV-2, while it was nearly equivalent for IBV. Additionally, we noted that the viral detection limits for IAV and RSV were superior in the automated system compared to the manual assays (lines 313–316).
Comment 4: Period of the study should be mentioned in the material methods
Response 4: We thank the reviewer for this suggestion. The period of the study has now been included (line 108).
Comment 5: Results of the negative samples need to show the specificity of the developed methods. What is the sensitivity and specificity of the developed methods? They need to be mentioned and also compared with the manual method.
Response 5: We thank the reviewer for this suggestion. We confirmed that virus-free saliva and nasal swab solutions were negative for SARS-CoV-2, IAV, IBV, and RSV genes using the geneLEAD/VIASURE indicating its high specificity as noted in lines 128–130, 240–243, and 316–319. Our study used virus-free saliva and nasal swab solution as negative controls or virus-spiked samples with known viral titers and expected Ct values; thus, we believe calculating these performance metrics (sensitivity and specificity) may be entirely inappropriate. We suggest the need for future studies using co-infected clinical samples with unknown viral titers, as noted in the study limitation (lines 366–369).
Comment 6: How the quantitation of the samples that were spiked was quantified for the specific virus needs to be described.
Response 6: We thank the reviewer for this suggestion. Quantification was performed by evaluating the viral titer of the used viruses. The methods for virus propagation and viral titer evaluation, previously described in the supplementary materials, have now been moved to the main text (lines 87–108).
Round 2
Reviewer 2 Report (Previous Reviewer 2)
Comments and Suggestions for Authors
While the authors have adequately responded to all of my comments and suggestions, I still believe that the major flaws pointed out in my initial review remain unresolved. Nevertheless, aside from these two issues, the rest of the work is appropriate for publication.
Author Response
We are greatly indebted to the reviewers for their thorough review of our manuscript and for providing us with insightful comments and suggestions. Subsequently, we have attentively revised the manuscript and highlighted all the changes in red. In addition, our point-to-point responses to the reviewers’ comments are provided below:
Reviewer 2:
Comments and Suggestions for Authors
While the authors have adequately responded to all of my comments and suggestions, I still believe that the major flaws pointed out in my initial review remain unresolved. Nevertheless, aside from these two issues, the rest of the work is appropriate for publication.
Response: We thank the reviewer for the feedback and acknowledge the limitations he/she has pointed out. We have described these limitations in our manuscript (lines 323–326 and 365–375) to avoid over interpretation of the results by the readers. We hope to test the applicability of geneLEAD/VIASURE using clinical samples in the future.

Reviewer 3 Report (Previous Reviewer 1)
Comments and Suggestions for Authors
The manuscript indicates (line 108) that the study took place from April 2023 to April 2025. Nonetheless, the ethical approval is dated March 2025, which presents a significant ethical issue. Beginning or carrying out a study that includes human participants (or sensitive data) without first securing ethical approval breaches established research ethics and institutional guidelines. The authors need to address this inconsistency and present proof that the study commenced only after obtaining the necessary ethical approval.
Author Response
We are greatly indebted to the reviewers for their thorough review of our manuscript and for providing us with insightful comments and suggestions. Subsequently, we have attentively revised the manuscript and highlighted all the changes in red. In addition, our point-to-point responses to the reviewers’ comments are provided below:
Reviewer 3:
Comments and Suggestions for Authors
The manuscript indicates (line 108) that the study took place from April 2023 to April 2025. Nonetheless, the ethical approval is dated March 2025, which presents a significant ethical issue. Beginning or carrying out a study that includes human participants (or sensitive data) without first securing ethical approval breaches established research ethics and institutional guidelines. The authors need to address this inconsistency and present proof that the study commenced only after obtaining the necessary ethical approval.
Response: We appreciate the reviewer’s detailed confirmation of the ethics approval date. Initially, all our experiments were conducted using commercially available saliva samples as mentioned in the manuscript. Therefore, ethical approvals for these samples were not required. However, one of the reviewers in the initial peer review report dated January 15, 2025, suggested further experiments using additional samples other than saliva. Therefore, we sought ethical approval only for the additional experiments involving the collection and use of nasal swabs from volunteers in our laboratory, which were conducted in April 2025 after obtaining the approval on March 26, 2025. This information has been described in lines 124–126 and 394–397 in the manuscript.
Round 3
Reviewer 3 Report (Previous Reviewer 1)
Comments and Suggestions for Authors
Accepted
This manuscript is a resubmission of an earlier submission. The following is a list of the peer review reports and author responses from that submission.
Round 1
Reviewer 1 Report
Comments and Suggestions for Authors
The manuscript entitled “Performance Evaluation of a Fully Automated Molecular Diagnostic System for Multiplex Detection of SARS-CoV-2, Influenza A/B Viruses, and Respiratory Syncytial Virus” by James G. Komu et al focused on the detection of respiratory injections using the multiplexed assay. However, the study presents several limitations, including
The study used only 2-4 samples spiked samples, which is insufficient for validating any assay and also lacks clinical validation of the study.
The study also lacks a comprehensive understanding of emerging variants of SARS-CoV-2, which could impact the system's sensitivity and specificity.
The comparison between geneLEAD/VIASURE and manual assays focuses on Ct values without delving into broader performance metrics, such as sensitivity, specificity, or clinical accuracy.
The multiplexing capability of the system is not rigorously assessed, with no discussion of potential interference between simultaneous targets or the impact of mixed infections with varying viral loads
Is there no extension step in the PCR cycle (in the line no. 109-111).
Reviewer 2 Report
Comments and Suggestions for Authors
In the manuscript with No. Diagnostics-3413502, the authors evaluated the performance of a fully automated sample-to-result platform by comparing the platform with manual assays to detect the four viruses in the viruses spiked samples. The results indicated good performance of the automated platform. There are following comments and suggestions.
1. The performances of the manual assays of IAV/IBV and RSV were not good due to the sequences of the adopted primer/probes not suitable for general viruses, as indicated in Table 2. Therefore, for evaluating the automated platform, well-designed primer/probe sets should be used as manual assays. This is a limitation of the study.
2. In Line 114, "Cycle threshold (Ct) values of 113 ≤40 for the four virus targets and either a Ct value of ≤40 or no signal for the IC were 114 considered a valid positive result." Why did the author think no signal for the IC is acceptable for positive results?
3. In Line 130-131, the cutoff Ct values are different for the four viruses. Are there any references to support it?
Reviewer 3 Report
Comments and Suggestions for Authors
The article is of little scientific/disciplinary interest. Although it is adequately structured, I have concerns about it that prevent me from accepting this work:
1) Why use saliva for the validation of this diagnostic system, excuse me, of this manual real time kit? As good as saliva is as a sample, the collection of this matrix is not yet standardised with precise guidelines. Moreover, the sample of choice for the viruses analysed is undoubtedly the nasopharyngeal swab.
2) The general problem with this work is the validated kit itself: why use a manual diagnostic system, which also involves extracting the sample, when there are so many other diagnostic systems today that can detect many more viruses at the same time, without extraction?
3) for a validation article of this type, 28 references are very few.
Author Response
The article is of little scientific/disciplinary interest. Although it is adequately structured, I have concerns about it that prevent me from accepting this work:
Comment 1:
Why use saliva for the validation of this diagnostic system, excuse me, of this manual real time kit? As good as saliva is as a sample, the collection of this matrix is not yet standardized with precise guidelines. Moreover, the sample of choice for the viruses analyzed is undoubtedly the nasopharyngeal swab.
Response to comment 1:
As mentioned in lines 57–59, we agree with the reviewer that nasopharyngeal samples (NPS) are the gold standard for diagnosing respiratory viruses. However, saliva samples have also been used to diagnose SARS-CoV-2, influenza viruses, and RSV with comparable outcomes to NPS samples. Nevertheless, based on the reviewer’s suggestion, we were willing to proceed with using NPS samples for comparison purposes. Unfortunately, the ethical approval for the current study only allowed us to collect nasal swab samples. Therefore, we performed additional experiments using pooled nasal swab solutions. We added information regarding the collection of the nasal swab solutions from two volunteers in our lab in lines 104–112. Subsequently, the results have been summarized in Table 2 and lines 215–221. A discussion of these results has been provided in lines 303–311.
Comment 2:
The general problem with this work is the validated kit itself: why use a manual diagnostic system, which also involves extracting the sample, when there are so many other diagnostic systems today that can detect many more viruses at the same time, without extraction?
Response to comment 2:
We thank the reviewer for highlighting this issue. While it is true that many automated sample-to-result platforms are available for the multiplexed diagnosis of viruses, they are not readily accessible for routine diagnosis in many clinical laboratories. As such, our study aimed to compare geneLEAD/VIASURE with the manual assays that involve manual extraction and RT-qPCR separately. However, we agree that it might be necessary to compare the geneLEAD/VIASURE with other similar sample-to-result platforms in the future.
Comment 3:
For a validation article of this type, 28 references are very few.
Response to comment 3:
We thank the reviewer for raising this concern. We critically reviewed many references and specifically included those that we felt would enhance the quality of our manuscript. Nevertheless, in response to the reviewer’s suggestions, we added a few more references to our manuscript, increasing the total number to 33. Furthermore, our supplementary materials contains an additional 3 references on the methodology used to prepare the virus stocks. Thus, the total number of references in our manuscript and supplementary materials is now 36.